# The effects of problem-based learning (PBL) on undergraduate medical students' critical thinking and communication skills development: A scoping review across resource-rich and resource-limited settings (2015–2024)

**Bruce Ayabilla Abugri**[ID][1], **Maxwell Ateni Assibi**[1], **Anthony Amalba**[2], **Patience Kanyiri Gaa**[3], **Sophia E.A. Kpebu**[1], **Patience Afua Adwaapa Karikari**[1], **Victor Mogre**[1]*

**1** Department of Health Professions Education and Innovative Learning, School of Medicine, University for Development Studies, Tamale, Ghana, **2** Department of Pharmacy Practice, School of Pharmacy and Pharmaceutical Sciences, University for Development Studies, Tamale, Ghana, **3** Department of Dietetics, School of Allied Health Sciences, University for Development Studies, Tamale, Ghana

* vmogre@uds.edu.gh

## Abstract

### Background

Problem-based learning (PBL) is widely recognized as an effective method for enhancing critical thinking and communication skills in medical education. However, in first-year medical students, the specific and detailed effects have yet to be explored, especially across resource-rich and resource-limited educational settings.

### Objective

The scoping review evaluates the effects of problem-based learning on the development of critical thinking and communication skills of first-year medical students and compares the effectiveness of the method in resource-rich and resource-limited environments.

### Methods

This scoping review followed Arksey and O'Malley's framework. A systematic search of the literature was conducted on Google Scholar, PubMed, and ProQuest for studies published between 2015 and 2024. Included studies were screened according to the defined criteria, and data were extracted on study characteristics, interventions, and outcomes measured.

**Data availability statement:** All relevant data are within the manuscript and its Supporting Information files.

**Funding:** The author(s) received no specific funding for this work.

**Competing interests:** The authors have declared that no competing interests exist.

## Results

A total of eight studies were found that met inclusion criteria: only one among them exclusively studied first-year students. The results of this cross-sectional study (with a descriptive self-report questionnaire) found that first-year students have a positive attitude toward PBL since they reported the following: high engagement (97%) and improved teamwork (87%). The included studies reported that PBL was associated with perceived improvements in critical thinking and communication skills among undergraduate medical students in both resource-limited and resource-rich educational environment. There were methodological variations, and some regions were entirely unrepresented. All the reviewed studies were published in Asia, with no studies from Africa.

## Conclusion

The reviewed studies suggest that PBL may be associated with perceived improvements in critical thinking and communication skills; however, findings remain tentative due to methodological and contextual limitations.

## 1. Introduction

Medical education has continually adapted to meet the shifting expectations of modern healthcare and the preparation of competent, reflective practitioners. Harden [1] describes medical education as a lifelong pathway that moves seamlessly from undergraduate studies to postgraduate specialization and, ultimately, to continuing professional growth. While this structure has provided stability and coherence, the manner in which medicine is taught has not remained static. Over the past century, education in the health professions has evolved through several waves of reform. The early apprenticeship model where learning depended heavily on imitation and bedside observation gradually gave way to the scientifically grounded curriculum introduced through the 1910 Flexner Report. That reform set lasting international standards for academic rigor and institutional accountability in medical training [2]. Yet, in focusing so intensely on scientific precision, it inadvertently de-emphasized the development of softer but equally vital skills such as clinical reasoning, empathy, and communication capacities that are now widely recognized as central to competent practice [3].

In the decades that followed, medical education entered yet another period of transformation. The growing presence of digital tools and interactive technologies virtual simulations, immersive virtual reality, and computer-based scenarios has opened new possibilities for both access and engagement in learning [4,5]. At the same time, competency-based education began to reframe assessment, shifting attention from how long a student spends in training to what that student can actually do [6]. Within this dynamic context, problem-based learning (PBL) emerged as one of the most influential innovations. Grounded in authentic clinical

cases, it invites learners to think critically, collaborate, and take responsibility for their own learning [2,7]. Introduced at McMaster University in 1969 [8], PBL has since spread across continents, including its adoption in Ghanaian medical schools in the year, 2007 [9,10] and has reoriented medical teaching from the authority of the lecturer toward the active engagement of the student [11,12].

Nevertheless, today's learners encounter a learning environment unlike that of previous generations. The move from traditional, lecture-heavy courses to the interactive and self-directed demands of PBL can be disorienting especially for first-year medical students still adjusting to university life. These challenges are more pronounced in settings with limited resources, where access to technology, infrastructure, and experienced facilitators is restricted. By contrast, institutions in resource-rich contexts can rely on advanced digital tools and well-trained tutors to support students [13]. This uneven landscape raises important questions: does PBL work equally well across such diverse environments, and to what extent does access to technology influence its outcomes.

Despite these contextual differences, a growing body of evidence continues to affirm the educational value of PBL. Reviews consistently highlight its contribution to analytical reasoning, problem-solving, and communication development [14,15]. For instance, a study in Pakistan observed that roughly sixty percent of students perceived improvement in their critical thinking after PBL exposure [16], and subsequent systematic reviews affirm its broader pedagogical benefits [17]. Research conducted in Lebanon further revealed that team-based learning, a related pedagogical form produces sustained enhancements in communication competence [18].

To ensure conceptual precision, this review applies established definitions of its two central constructs. Critical thinking in medical education denotes the ability to interpret, analyze, and apply information judiciously in clinical decisions [19,20]. Empirical studies frequently operationalize it through measures such as analytical reasoning, problem-solving and collaborative skills [21–23]. Communication skills are understood as the capacity to convey, receive, and interpret messages effectively within clinical and collaborative settings [24,25]. These skills are often examined through dimensions such as teamwork, interpersonal exchange, and presentation ability. This review interprets the available evidence through these operational indicators while retaining their overarching conceptual meaning.

## 1.1 Rationale

Despite a plethora of studies on Problem-Based Learning (PBL) in medical education, gaps remain in the literature. While PBL is acknowledged for nurturing communication and critical thinking skills, few studies have laid bare its impact on first-year medical students: just as few have done so across different learning contexts-from resource-rich to resource-constrained environments. Moreover, it remains to be seen whether first-year medical students transitioning to self-directed learning face any specific challenges or gains in terms of communication and critical thinking skills utilization by using PBL. Addressing this knowledge gap would help optimize PBL's implementation and ensure effectiveness among different groups of learners. In this scoping review, the authors summarized data that describe and evaluate PBL efficacy in both resource-rich and resource-constrained settings and identified crucial aspects in further investigations to chart the way for future research. This study lays the groundwork for future investigations into how first-year medical students adjust to PBL and whether more instructional support is required to optimize its effects on their learning outcomes such as critical thinking and communication skills.

## 1.2 Working definition of problem-based learning (PBL)

In this review, we define problem-based learning (PBL) as a student-centred approach delivered through facilitated small-group tutorials in which an authentic problem scenario (the trigger) is presented first and serves as the main organiser of learning. With guidance from a tutor who facilitates process and discussion rather than delivers content, students analyse the trigger, identify what they know and what they need to learn, formulate learning issues as objectives, complete self-directed study, and reconvene to integrate, discuss, and apply new knowledge. What sets PBL apart from other

active-learning approaches is that students generate the learning agenda from the problem and use a structured tutorial cycle to drive learning; in case-based learning, cases more often support application or consolidation of content that has already been taught and objectives tend to be more instructor-defined, while inquiry-based learning is organised around learner questions or investigations that may not use a problem trigger as the central curriculum organiser or follow the PBL tutorial cycle [26].

### 1.3 Objective

This scoping review maps available evidence on the effects of PBL on critical thinking and communication skills in undergraduate medical education, with a focus on first-year medical students. It seeks to identify key themes and gaps in the literature and compare PBL's effectiveness in resource-rich and resource-limited settings.

### 1.4 Research questions

1. What is the effect of PBL on critical thinking skills among undergraduate medical students, particularly first-year students?

2. How does PBL influence the development of communication skills among undergraduate medical students, including first-year students?

3. How do PBL outcomes (critical thinking and communication skills) differ between resource-rich and resource-limited settings for undergraduate medical students?

## 2. Method

### 2.1 Protocol and registration

The review followed Arksey and O'Malley's Framework for conducting scoping reviews which consists of five main stages. The framework consists of stages such as 1) identifying the research question; 2) ascertaining relevant studies; 3) determining study selection; 4) charting the data; and 5) collating, summarizing and reporting the results [27]. We presented our review report using the PRISMA Extension for Scoping Reviews [28] as shown in Fig 1. A protocol was developed but not formally registered.

### 2.2 Inclusion and exclusion criteria

The inclusion criteria were that: 1) the main research topic should be related to Problem-Based Learning methodology; 2) the participants should be undergraduate medical students; 3) the main outcomes measured included one or more of the following: critical thinking skills, communication skills, or generic skills problem-solving, analytical reasoning, and collaborative skills; 4) studies should have been published in peer-reviewed journals between 2015 and 2024; The review included empirical studies such as experimental, quasi-experimental, and observational studies; descriptive papers; and studies that used qualitative, quantitative, or mixed-methods approaches. The review excluded the following: 1) studies published in languages other than English, 2) conference papers, 3) grey literature, including opinion articles, commentaries, and editorials, 4) systematic reviews, and 5) scoping reviews. Practical issues, such as a lack of resources to translate non-English studies or the research team not possessing enough proficiency in languages other than English, constrained the review to English language publications. Access and identification of such studies may be difficult as these are often not indexed in databases of the commonly searched. While this method potentially introduces a language bias, this was done to ensure the feasibility and manageability of the review.

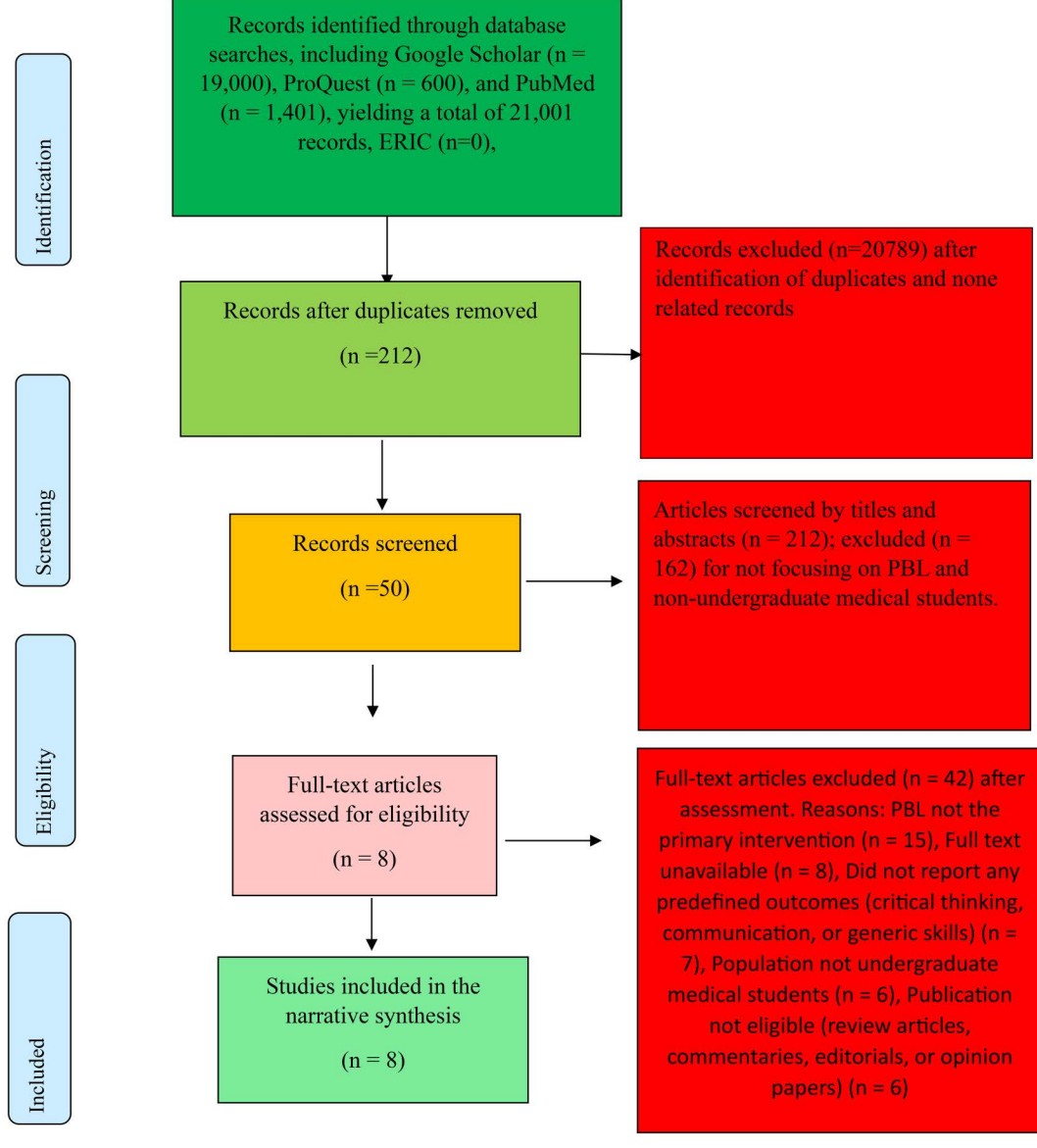

**Fig 1. PRISMA-ScR flow diagram.**

## 2.3 Information sources

We utilized multiple electronic databases to identify relevant literature to meet the objectives of the scoping review. Google Scholar, PubMed, ProQuest, ERIC and the reference lists of included studies, were searched for related literature. These databases were selected to ensure complete coverage of studies related to problem-based learning, Critical thinking, Communication and generic skills (Problem-solving and analytical reasoning). There was no restriction on the geographical location of studies or study design for inclusiveness. We applied a snowballing approach, reviewing reference lists of included studies to identify additional relevant sources.

## 2.4 Search strategy

A structured search strategy was developed and subsequently run on Google Scholar, PubMed, ProQuest, ERIC and the reference lists of included studies following the establishment of the review objectives. The search was conducted using a structured approach that includes MeSH terms combined with Boolean operators like AND, OR, and NOT to enhance precision and relevance in identifying the pertinent literature.

The main search terms used were "Problem-Based Learning" AND "Critical Thinking Skills", "Problem-Based Learning" AND "Problem-Solving Skills" OR "Communication Skills", "Problem-Based Learning" AND "Medical Education", "Problem-Based Learning" AND "Undergraduate Medical Students", and "Problem-Based Learning" AND "Communication Skills". There were no restrictions regarding the date of publication, study design, or country of origin to ensure a comprehensive and inclusive review of the available evidence.

The search was conducted from October 9, 2024, to February 7, 2025. All the retrieved citations were imported into EndNote, version 20.0, for title and abstract screening and data characterization to facilitate systematic organization and analysis. There were no restrictions regarding study designs and country of origin. This enabled a comprehensive and inclusive review of the available evidence.

## 2.5 Selection of a source of evidence

Studies underwent title and abstract screening, followed by full-text review by two independent reviewers (BAA & AMA). Disagreements were resolved through discussion. A PRISMA flow diagram (Fig 1) illustrates the selection process. Where uncertainty regarding study eligibility arose, all authors reviewed the study collaboratively and deliberated on its alignment with the inclusion criteria to arrive at a consensus. If necessary, discussions between all authors took place (BAA, AA, PG, SK, AM and VM) to resolve such uncertainties.

## 2.6 Charting

Collaboratively, a data extraction table was adapted from the previous study by the research team members to ensure consistency and accuracy. The data extracted from the publications included general details such as the year, author, study objectives, and study location, as well as specifics about sample size, study design, methodology, intervention, duration, and measured outcomes. The extracted data were systematically compiled into one spreadsheet in Microsoft Excel after which coding and analysis were carried out.

## 2.7 Critical appraisal of individual sources of evidence

For this scoping review, no critical appraisal was done because the study seeks to map out evidence instead of the quality of individual studies.

## 2.8. Synthesis of results

**2.8.1. Thematic synthesis of findings.** The data description was summarized descriptively, using mainly thematic trends, and contextual comparison between resource-rich and poor settings. To strengthen rigor, we conducted a descriptive thematic synthesis of the included studies, drawing on Braun and Clarke [29] framework and guided by Nowell, Norris [30] to ensure trustworthiness. This process enabled a systematic thematic analysis of the eight included studies, which yielded five key themes: methodological insights and sample size, critical thinking skills, the interconnectedness of critical thinking and communication skills, first-year medical students, and resource-rich versus resource-limited settings.

## 3. Results

### 3.1. Selection of sources of evidence

The systematic literature search resulted in the identification of 21,001 articles from Google Scholar, PubMed, and Pro-Quest. After removing duplicates and unrelated articles, 212 unique articles remained for title and abstract screening by two independent reviewers (BAA & AMA). This initial screening excluded a vast majority of studies for not meeting the predefined inclusion criteria. The primary reasons for exclusion at this stage were: (1) studies conducted on populations other than undergraduate medical students (e.g., nursing, allied health, or postgraduate students); (2) interventions that were not Problem-Based Learning (e.g., traditional lectures, team-based learning without a PBL foundation); (3) outcomes that did not include critical thinking, communication, or related generic skills; (4) publication types outside the scope of peer-reviewed empirical studies (e.g., editorials, commentaries, conference papers, systematic reviews); and (5) publication dates outside the 2015–2024 timeframe.

Following the identification process, 212 records were screened by titles and abstracts. During this stage, 162 articles were excluded for not aligning with the inclusion criteria. Specifically, studies were excluded for not focusing on problem-based learning (PBL) (n = 58), addressing outcomes outside the review scope (n = 28), involving non-undergraduate populations (n = 24), representing ineligible publication types such as reviews or commentaries (n = 20), or lacking accessible abstracts for preliminary assessment (n = 32).

Of the 50 studies assessed for eligibility, eight met all inclusion criteria and were retained for the final review. As shown in Fig 1, 42 studies were excluded for reasons including lack of focus on problem-based learning (PBL), unavailability of full texts, absence of relevant outcomes, non-undergraduate populations, or ineligible publication types. Data were charted using a predefined data extraction sheet developed a priori. The sheet captured author and year, country, study design, setting, population and sample size, intervention or exposure and outcome measures (As shown in Table 1). This systematic process ensured transparency, rigor, and reproducibility of the review's findings.

### 3.2. Synthesis of result

**3.2.1. Characteristics of the included studies.** All the reviewed studies originated from Asia, with most (25%) each coming from China, Pakistan and India. Most of the studies (50%) were published in 2018. All the studies (100.0%) were cross-sectional. Also, majority (87.5%) of the studies had no comparator. Further details on the characteristics of the included publications are provided in Table 2.

**3.1.2. Methodological insights and sample size.** The reviewed studies employed diverse assessment methodologies to evaluate the effects of PBL on critical thinking and communication skills. These included self-reported questionnaires, descriptive feedback analyses, and validated assessment instruments. The heterogeneity in data collection methods complicates direct comparisons across studies and limits the generalizability of findings. The lack of standardized assessment tools and variations in study designs further emphasize the need for consistency in future research. The reviewed studies varied significantly in sample sizes, ranging from 52 to 464 participants, with a median of 149.

**3.1.3. Critical thinking skills.** Critical thinking was the primary focus of four studies [16,31–33]. These studies consistently demonstrated that PBL enhances analytical reasoning and problem-solving abilities by engaging students in structured, step-by-step strategies to solve complex clinical problems. The findings suggest that PBL encourages deeper cognitive processing, enabling students to develop essential reasoning skills that maybe necessary for clinical decision-making.

**3.1.4. The interconnectedness of critical thinking and communication skills.** While none of the studies focused exclusively on communication skills, findings from multiple the studies reviewed suggest that PBL fosters teamwork, interpersonal communication, and presentation abilities. These skills are crucial for effective patient interactions and

**Table 1.** Data extraction sheet.

| Author, Year | Study Location | Study objective | Study design | Sample Size and Participants | Inter-vention | Duration | Outcomes |
|---|---|---|---|---|---|---|---|
| Asad et al, (2015) | Pakistan. | To assess the effectiveness of PBL in fostering problem-solving and critical reasoning skills among medical students. | Cross-sectional | 193, first and second-year medical students | PBL | 1 Month | Critical thinking & Problem-solving Skills |
| Al-Shaikh, (2015) | Saudi Arabia | To evaluate medical students' perceptions of PBL. | Descriptive cross-sectional | 52, second-year medical students | PBL | 4weeks, | Self-learning, critical thinking, |
| S. Hande et al, (2015) | India | To investigate students' perceptions of PBL's role in acquiring knowledge, skills, and attitudes. | Cross-sectional study | 464 undergraduate medical students | PBL | 6 years | Generic Skills (analytical reasoning, problem-solving and collaborative skills) |
| Mughal et al, (2018) | Pakistan. | To assess collaborative problem-solving (CPS) skills in undergraduate medical students during PBL sessions. | Analytical comparative cross-sectional study | 210, first, second- and third-year medical Students | PBL | 6 Months | Level of CPS (Communication and Critical thinking skills |
| Latif et al, (2018) | Saudi Arabia | To compare the effectiveness of role-play and debate sessions in enhancing communication and critical thinking skills in PBL. | Comparative, cross-sectional, questionnaire-based study | 185 second-year undergraduate female medical students | PBL | One academic year | Improving critical thinking and communication skills |
| Yadav et al, (2018) | Nepal | To assess attitudes and perceptions of medical students toward PBL. | Descriptive study using a self-administered questionnaire | 113 first-year MBBS students at Chitwan Medical | PBL | None | Effectiveness in improving problem-solving skills, communication, |
| He et al, (2018) | China | Compare the effectiveness of PBL and lecture-based learning (LBL) in improving medical students' problem-solving abilities using questioning strategies | Comparative study | 104 medical students. | PBL | None | Improvement in Problem-Solving Abilities (critical thinking) |
| Pu et al, (2019) | China | To explore the relationship between critical thinking (CT) disposition and students' performance in problem-based learning (PBL). | Analytical cross-sectional study | 102, third-year undergraduate medical students | PBL | One semester | Critical thinking disposition |

collaboration in medical practice [34,35]. The four remaining studies [34–37] examined both critical thinking and communication skills, highlighting their interdependence. The results suggest that PBL fosters both cognitive and interpersonal skills simultaneously by engaging students in collaborative problem-solving. Through group discussions and structured interactions, students develop the ability to articulate their reasoning effectively, negotiate solutions, and refine their communication.

**3.1.5. First-year medical students.** Of all the reviewed studies, only one by Yadav, Piryani [36] was conducted with a focus on first-year medical students in Nepal, and it found that these students generally viewed Problem-Based Learning positively, seeing PBL as especially important for enhancing critical thinking and communication skills. Of the different aspects of PBL, the students expressed very high opinion results: engagement (97%), relevance to medical education (89%), and perceived merits of cooperation in learning (87%), with other efficacy percentages for self-directed learning high at 89.4% and improved study habits (84%), where only (50%) expressed opinions of high time consumption toward PBL. An additional role of the tutor was highly significant, in this respect, mentioned by (80.5%) for assistance in discussions and skills development. The findings above show that although PBL does foster the development of critical thinking and communication, this finding may not be attributed to PBL effect hence the study used self-reported questionnaire and a cross-sectional design which cannot establish causality. The rest of the studies reviewed combined undergraduate medical students (first, second and third years), without any focus on the experiences of first-year students

**Table 2. Characteristics of publications included in the review.**

| Characteristics of publication | Number (Percentage) |
|---|---|
| Year of publication | |
| 2015 | 3(37.5%) |
| 2018 | 4(50.0%) |
| 2019 | 1(12.5%) |
| Geographic distribution of the publications | |
| China | 2(25.0%) |
| Pakistan | 2(25.0%) |
| India | 2(25.0%) |
| Nepal | 1(12.5%) |
| Saudi Arabia | 1(12.5%) |
| Study population | |
| Undergraduate medical students | 8(100.0%) |
| Study design | |
| Cross-sectional studies | 7(87.5%) |
| Comparative studies | 1(12.5%) |
| Comparator | |
| With Comparison | 1(12.5%) |
| Without Comparison | 7(87.5%) |

regarding the effects of PBL on developing their critical thinking and communication skills. It is hence difficult to ascertain whether first-year students face unique difficulties adjusting to PBL in a way that further affects the development of critical thinking and communication skills among this specific cohort. Future studies should approach PBL support specifically to study the first-year medical students' learning adaptation, self-directed learning development, and critical thinking and communication skills achievements, notably in resource-constricted settings, applying more insightful experimental or longitudinal designs.

**3.1.6. Resource-rich vs. resource-limited settings.** The reviewed studies highlight differences in PBL's implementation across resource-rich and resource-limited educational settings. Studies conducted in well-resourced environments such as Saudi Arabia and China, where high-quality infrastructure and experienced faculty were available [31–34] reported substantial improvements in both critical thinking and communication skills. The structured nature of PBL in these settings, often supported by digital tools and expert facilitators, enhances student engagement and learning outcomes.

In contrast, studies from resource-limited settings such as Pakistan, India, and Nepal [16,35–37] suggest that PBL remains effective in developing critical thinking and communication skills even when institutional resources are constrained. The collaborative and discussion-based nature of PBL appears to compensate for infrastructural limitations by fostering active learning and peer engagement. The findings indicate that despite contextual challenges, PBL can still provide meaningful educational benefits if appropriately adapted to resource-limited environments.

**3.1.7. Summary of evidence.** This scoping review highlights the effectiveness of Problem-Based Learning (PBL) in fostering critical thinking and communication skills among undergraduate medical students across both resource-rich and resource-limited settings. PBL enhances higher-order cognitive processes, such as analytical reasoning, problem-solving, interpersonal, collaborative skills which are essential for clinical decision-making. Notably, a research gap remains regarding the effects of PBL on first-year undergraduate medical students critical thinking

and communication development skills due to the adaptation challenges they face during introduction to PBL. Only study that has examined the first-year students' perception of PBL. While first-year medical students are included in one study, their learning difficulties and instructional needs are not given much attention. The first-year students' adaptability to PBL may either negatively or positively impact their development of critical thinking and these two core competencies, undoubtedly, are the key competencies in medical practice. It is important to note that the first-year medical students are mainly new to PBL, having been transitioning from a traditionally lecture-based instruction system, hence they may have some development issues surrounding independent learning, collaborative decision-making, and even the accomplishment of unstructured learning environments [35,38]. Such hurdles maybe more noticeable in low-resource settings where limited access to faculty, technology, and even structured communication training could adversely affect their adaptation. Future studies ought to focus on conducting multi-centre studies with more standardized methodologies, which will lend to the reliable comparison of findings while at the same time looking into the contextual adaptation of PBL in low-resource settings. Equally exigent is the expansion of research in underrepresented regions such as Africa to understand how PBL can effectively work across cultures with diverse educational backgrounds. Structured approaches to assessing programs and technology-enhanced PBL models should be given the highest priority, to enhance the engagement and efficacy of PBL while ensuring equal access to quality medical education around the globe [36,37].

## 4. Discussion

This scoping review mapped evidence from eight studies on the effects of PBL on critical thinking and communication skills. Our thematic analysis reveals a complex but largely positive picture, suggesting PBL is associated with the development of these essential competencies, while also highlighting critical gaps in the research, particularly concerning first-year students and methodological rigor.

### 4.1. Methodological insights and sample size

The reviewed studies revealed substantial methodological heterogeneity in assessing PBL's influence on critical thinking and communication skills. Researchers employed diverse evaluation tools, including self-reported questionnaires, descriptive feedback analyses, and validated psychometric instruments. This diversity, while reflecting innovation in assessment design, also complicates cross-study comparability and limits the generalizability of results. Recent reviews confirm that the absence of standardized tools and inconsistent methodological rigor remain persistent barriers to synthesizing PBL outcomes [39]. Sample sizes among the reviewed studies ranged from 52 to 464 participants, with a median of 149, underscoring wide variability in statistical power and representativeness. Meta-analytic evidence similarly notes that many PBL investigations rely on small or convenience samples, which constrain the robustness of inferences and magnify study-level heterogeneity [40].

Despite these methodological inconsistencies, the reviewed literature continues to highlight PBL's promise as an instructional approach that enhances learners' higher-order reasoning, collaboration, and reflective communication [39]. Future research should prioritize larger, longitudinal, and multi-institutional studies using validated and standardized assessment.

### 4.2. Critical thinking

Almost all the studies reviewed suggest consistent gains in students' analytical reasoning, problem-solving, self-directed learning, teamwork, interpersonal interaction, and presentation skills. PBL provide students with the opportunity to interact with case-based scenarios, which fosters higher-order cognitive and communication processes like analytical reasoning, problem-solving, collaborative and interpersonal skills, which are fundamental for clinical decision-making [31,32]. The interactive components of PBL, including group discussions and role-playing, have a better reinforcing effect on critical

thinking and communication skills by enabling students to express their reasoning, defend their decisions, and take part in peer learning.

## 4.3. The interconnectedness of critical thinking and communication skills

Reviewed studies from China and Saudi Arabia indicates that PBL interventions have made a significant contribution to the growing body of evidence concerning the ability of PBL in promoting critical thinking and communication skills [31,33]. Besides fostering critical thinking, PBL has been reported to associated with the indicators of improved communication skills like teamwork, interpersonal communication, and presentation skills [35,37].

## 4.4. Resource-rich vs resource-limited settings

Across the reviewed studies, PBL appeared feasible in resource-limited settings where formal communication-skills training and extensive instructional infrastructure may be limited. In these contexts, the small-group tutorial cycle can support verbal expression, active listening, and peer feedback through repeated discussion and reflection, with the tutor facilitating group process rather than delivering content [36]. Consistent with this, peer learning and structured case discussions may partially substitute for faculty-led instruction when faculty time, teaching space, or learning resources are constrained [35,36].

In contrast, studies conducted in settings like Saudi Arabia [34] and China [32,33] with stronger infrastructure and faculty capacity more often reported the use of additional structured learning activities (for example, role-plays, debates, or expert-led sessions) to extend practice opportunities for communication and reasoning. In the reviewed evidence set, these studies reported improvements in communication and reasoning outcomes, although it is not always clear whether these gains were attributable to PBL alone or to PBL combined with these supplementary activities. Where PBL is implemented with such enhancements, they should be reported explicitly as part of the intervention package to avoid misattributing effects to PBL alone.

Essential resources for PBL implementation include trained facilitators, protected time, small-group space, and access to learning materials that support self-directed study. In resource-limited contexts represented in this review (for example, Pakistan, India, and Nepal), constraints in faculty development and institutional capacity were common, yet PBL remained implementable through a structured tutorial cycle that emphasised peer learning and guided discussion. A minimal viable package in such settings includes contextually relevant problem triggers, basic learning resources, targeted tutor preparation, and peer-mentoring structures that reduce facilitator workload.

Overall, the reviewed evidence suggests that PBL can support critical thinking and communication skill development across resource contexts, but the magnitude of effects may depend on implementation quality and the availability of enabling resources [35,36].

## 4.5. First-year medical students

However, little is known about the acculturation of first-year students into the PBL approach, and few reports have documented it, especially in resource-limited settings. With first-year students facing significant challenges in the transition from traditional, teacher-center learning environments to the more self-directed and collaborative nature of PBL, challenges might include self-directed learning, time management, and group dynamics; all of which are crucial within PBL settings [13,16,31,35]. In resource-limited settings, however, the aforementioned challenges may be compounded by insufficient faculty support, inadequate technical resources, and limited training in communication skills. Due to the inherent requirements of PBL, first-year medical students may need additional assistance if they are to be successful in making the transition to this instructional approach. Explicit direction by curriculum committees within institutions, faculty development activities, or oversight by national accrediting agencies can help ensure a smoother transition. Such support is necessary to allowing students to achieve the maximum benefits of PBL, particularly in critical thinking and communications

skills acquisition. It's unknown how much they find it difficult to learn on their own, solve problems in groups, and adjust to a less regimented style of instruction than traditional lectures. Guidance can be provided through national accreditation bodies, curriculum review boards, and faculty development programs.

## 4.6. Influence of cultural and diverse educational backgrounds

A key limitation of this scoping review is that, although "cultural background" and "educational background" were recognized as possible moderating factors, the studies reviewed did not examine them systematically. The absence of this analysis restricts how far the findings can be applied in real-world settings, especially where problem-based learning (PBL) is introduced to groups of students with diverse cultural and educational profiles. Evidence increasingly shows that such factors shape how learners engage with and benefit from PBL. For instance, Kebaetse, Griffiths [41] reported that sociocultural norms such as valuing silence as a sign of respect hindered both self-directed learning and group interaction in an African medical school's PBL program. Similarly, Nicolaou, Televantou [42] observed that students with biomedical backgrounds began PBL pharmacology sessions with clear performance advantages. However, these advantages did not necessarily translate into greater learning gains compared to their peers from non-biomedical pathways.

Broader scholarship reinforces these observations. Chan, Gondhalekar [43] found that in non-Western settings, students' perceptions of tutor authority and group dynamics were shaped by cultural values such as power distance and collectivism. Likewise, Leatemia, Susilo [44] showed that tutors' own cultural and educational orientations influenced their facilitation styles, which in turn affected how much autonomy students exercised. Taken together, these studies demonstrate that cultural and educational backgrounds are not peripheral issues; they fundamentally shape how PBL fosters communication, collaboration, and critical thinking. Addressing these moderators should therefore be a priority for future research.

## 4.7. Conclusion and future directions

Prioritizing multi-center studies and consistent techniques will help to boost future research. More comparison analyses and trustworthy results will be possible with the use of standardized evaluation instruments and consistent study designs. To ascertain whether first-year medical students need extra assistance with self-directed learning, adjusting to PBL, and cooperative problem-solving, more research should look into their experiences, especially regarding how their critical thinking and communication skills are developed due to the problem-based learning experience. This is especially crucial in environments with limited resources, as students might not have as many chances and opportunities in terms of resources, technology, faculty and infrastructure as compared to their counterparts in the developed world. By addressing these research gaps, PBL implementation techniques will be improved and universal access to high-quality medical education could also be improved. Furthermore, Future studies should focus on multi-Centre studies, standardized methodologies, and technology-enhanced PBL models to optimize its effectiveness in diverse educational contexts. This review found no studies from Africa, highlighting a gap in PBL research within low-resource settings. Future studies should explore how PBL is implemented in diverse cultural and economic contexts.

## 4.8. Limitations

There are several limitations to this scoping review. The overall number of studies that were included was small (n = 8), and only one of these focused specifically on first-year medical students, which lowers the generalizability and degree of analysis. The studies varied considerably in design, sample size, and assessment tools. Due to heterogeneity in study designs, populations, and outcomes, conducting a meta-analysis was not appropriate. This is consistent with the aims of scoping reviews, which focus on mapping the breadth of evidence rather than producing pooled effect estimates [27,28]. Instead, we used both descriptive and thematic synthesis to summarize study characteristics, identify recurrent patterns, and highlight evidence gaps. Most of the studies employed self-reported data that may yield subjective bias and

undermine the validity of the findings. Furthermore, all the included studies were conducted in Asian countries, and none of them were conducted in African, European, or American settings, thereby decreasing the geographical generalizability of the evidence. Lastly, the lack of longitudinal or experimental designs restricts the degree to which long-term impacts or causality may be inferred.

### 4.9. Conclusion

From the scoping review, PBL may be associated with perceived improvements of critical thinking and communication skills. The findings, however remain tentative, need to be drawn cautiously considering the limited number of qualifying studies and the prevalence of cross-sectional, self-reporting designs. Specifically, there was only one study that focused exclusively on first-year medical students, and therefore, it was not straightforward to ascertain their specific experience and challenges with PBL. Moreover, the absence of studies from African and other underrepresented parts of the world limits the generalizability of the findings across various learning environments. Standardization of methods in future research and broader geographical representation can give elaborate insights about the effects of PBL and therefore could inform its effective application in both resource-rich and resource-limited settings.

### 4.10. Practical implications for educators implementing PBL

This review highlights several actionable lessons for educators who apply problem-based learning (PBL) in undergraduate medical education. In resource-rich settings, the priority is to make full use of available infrastructure. Digital platforms and simulations can enhance realism, while skilled facilitators can move beyond managing groups to guiding higher-order reasoning and professional reflection. These environments also make it possible to monitor learning progress and provide targeted feedback that supports both cognitive and communication skill development.

In contrast, educators in resource-limited contexts must rely on strong pedagogy rather than technology. The design of clear, relevant clinical cases and effective facilitation become the main tools for promoting learning. Structured peer mentoring can reduce faculty workload, and community settings can serve as authentic learning environments that connect theory with local health realities.

Supporting first-year medical students requires additional care. Because they often struggle with self-direction and teamwork, PBL should begin with structured orientation sessions that introduce its goals and methods. Early cases should be guided and gradually increase in complexity as students gain confidence. Educators should also teach collaboration explicitly, rather than assuming that teamwork will develop naturally.

Finally, cultural and educational diversity demands contextual adaptation. Case scenarios and facilitation styles should reflect local values, communication norms, and health-system conditions. Forming balanced groups and assigning rotating roles can ensure equitable participation. Ultimately, effective PBL implementation depends on educators' capacity to adapt its principles to their resources, learner needs, and cultural realities while maintaining its core aim, developing reflective, competent, and collaborative practitioners.

### Supporting information

**S1 File. PRISMA-ScR checklist for the scoping review.**
(DOCX)

### Author contributions

**Conceptualization:** Bruce Ayabilla Abugri, Anthony Amalba, Victor Mogre.

**Formal analysis:** Bruce Ayabilla Abugri, Maxwell Ateni Assibi, Patience Afua Adwaapa Karikari.

**Investigation:** Bruce Ayabilla Abugri.

**Methodology:** Bruce Ayabilla Abugri, Maxwell Ateni Assibi.

**Resources:** Bruce Ayabilla Abugri, Patience Kanyiri Gaa.

**Supervision:** Patience Kanyiri Gaa, Anthony Amalba, Sophia E. A. Kpebu, Victor Mogre.

**Writing – original draft:** Bruce Ayabilla Abugri.

**Writing – review & editing:** Patience Kanyiri Gaa, Anthony Amalba, Sophia E. A. Kpebu, Maxwell Ateni Assibi, Patience Afua Adwaapa Karikari, Victor Mogre.

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
