## [Decision Letter · Decision Letter 0]

2 Jun 2025

Dear Dr. Abugri,

Thank you for submitting your manuscript to PLOS ONE. After careful consideration, we feel that it has merit but does not fully meet PLOS ONE’s publication criteria as it currently stands. Therefore, we invite you to submit a revised version of the manuscript that addresses the points raised during the review process.

Please go through the comments by reviewers #2 and #3. Please ignore Reviewer #1's comments.

Regarding Reviewer #2's comment about the small sample of final papers and your presented conclusions, I have two suggestions - feel free to choose one, but if you want to do the minimum required then please go with option #1.

(1) Limiting and qualifying your conclusions based on the fact you have a very small sample without ability to conduct thematic analysis or inferential statistics. Related to this, I suggest adding a sub-section to your 'Discussion' section, titled 'Limitations', which will outline the limitations of your study.

(2) Expanding your final sample by relaxing some of your inclusion/exclusion criteria, which might change your research scope/objective/questions somewhat (don't go too far with this of course). The advantage of this approach is that you will be able to say more about your final sample of papers and support more of what you have to say, potentially with thematic analysis and perhaps even inferential statistics. In short, your contribution will be greater. Related to this, I suggest adding a sub-section to your 'Discussion' section, titled 'Contribution', which will outline the contributions of your study.

We look forward to receiving your revised manuscript.

Kind regards,

Rea Lavi

Academic Editor

PLOS ONE

**Journal Requirements:**

1. When submitting your revision, we need you to address these additional requirements. Please ensure that your manuscript meets PLOS ONE's style requirements, including those for file naming. The PLOS ONE style templates can be found at https://journals.plos.org/plosone/s/file?id=wjVg/PLOSOne_formatting_sample_main_body.pdf and https://journals.plos.org/plosone/s/file?id=ba62/PLOSOne_formatting_sample_title_authors_affiliations.pdf 2. Please provide a complete Data Availability Statement in the submission form, ensuring you include all necessary access information or a reason for why you are unable to make your data freely accessible. If your research concerns only data provided within your submission, please write "All data are in the manuscript and/or supporting information files" as your Data Availability Statement. 3. When completing the data availability statement of the submission form, you indicated that you will make your data available on acceptance. We strongly recommend all authors decide on a data sharing plan before acceptance, as the process can be lengthy and hold up publication timelines. Please note that, though access restrictions are acceptable now, your entire data will need to be made freely accessible if your manuscript is accepted for publication. This policy applies to all data except where public deposition would breach compliance with the protocol approved by your research ethics board. If you are unable to adhere to our open data policy, please kindly revise your statement to explain your reasoning and we will seek the editor's input on an exemption. Please be assured that, once you have provided your new statement, the assessment of your exemption will not hold up the peer review process.

Reviewers' comments:

Reviewer's Responses to Questions

**Comments to the Author**

1. Is the manuscript technically sound, and do the data support the conclusions?

Reviewer #1: Yes

Reviewer #2: Yes

Reviewer #3: Partly

2. Has the statistical analysis been performed appropriately and rigorously?

Reviewer #1: I Don't Know

Reviewer #2: N/A

Reviewer #3: N/A

3. Have the authors made all data underlying the findings in their manuscript fully available?

Reviewer #1: Yes

Reviewer #2: Yes

Reviewer #3: No

4. Is the manuscript presented in an intelligible fashion and written in standard English?

Reviewer #1: Yes

Reviewer #2: Yes

Reviewer #3: Yes

**Reviewer #1:**  Providing a rationale discussion for the difference in results that was provided through such huge revision of such numbers of the study would have provided a new prospective for the paper other than it's provided version

**Reviewer #2:**  The article is well written and covers the objective well. The scoping review has followed standard procedure and has explained its findings well.

The few minor issues found are:

1. The full form of TBL is not explained.

2. Table 1 is out of view in PDF and only visible in downloaded word file. The formatting may need to be changed by authors or the editorial.

3. Few language issues were seen, eg. use of "cannot established".

Two major issues found are:

1. Flow chart looks a bit confusing. The second exclusion isn't given a reason.

2. References might need to be redone according to journal guidelines and definitely need to be seen one by one. There are some inconsistencies like all caps headings.

**Reviewer #3:**  PBL in undergraduate learning is an interesting and under-researched area, and I hope the authors will consider directions for a future iteration of this paper.

I have 3 concerns about this paper:

1. It isn’t clear to me why so few papers were judged to meet the criteria. As I mention below, I used the authors’ search terms on only Google Scholar (and not on the other databases they used) and found easily found a number of articles that fit their stated criteria, were in fact based on these criteria, and discussed them clearly. The selection methods need to be more carefully articulated so they are defensible, or possibly, the authors might consider returning to the search results.

2. The tiny sample of articles in this paper does not justify its conclusions. One article on first-year medical students can suggest a way to proceed, but in itself it is not definitive of anything. The article methods are, as the authors note, variable and include attitude studies, questionnaires, and perceptions. It is difficult to draw conclusions from self-report.

3. The use of the words trials, causation, and guaranteed indicates a need for rethinking the kind of literature they are examining, much of which, again, is self-report or perception-based. These more clinical words do not fit this territory.

Line by line comments:

ll. 30, 33: major conclusions drawn on the basis of a single study

l. 46: “lots of”: please indicate which strategies and arrive at a number

l. 47: “time immemorial” indicates back thousands of years

l. 54: citation does not fit the assertion

l. 61: Barrows said PBL was not a method and noted the wide variety of objectives

l. 68: augusted is not a word

l. 72: “monumental historical proportions” : avoid these vague assertions

l. 81: Organizational issue: this needs to be moved elsewhere. No link to previous paragraphs.

ll. 83-5: in Ghana or in general? Define “several curricula”

l. 90: suggest using Garrison (1992) and Newman et al (1995) in addition to Facione. They have metrics for analysis

l. 101-102: important to distinguish between self-report, which is notoriously unreliable, and assessment.

l. 105: Organization issue: this is a new topic

L. 115: TBL is distinct and its use here does not seem to serve a purpose

l. 130-31: is this clearly done later?

l. 163-64: Where was this presented?

l. 224-25: rich and poor settings only thematic trends?

l. 235-36: please identify the “rigorous process.” This is important because so few studies were identified, and it presents a blank space in the process. Why were so few identified? I conducted a search using only 1 set of the search terms in Google Scholar and found a substantial number of studies that met the criteria set in this paper.

l. 242: showcases: find better word.

l. 310-13: while on its own this statement seems defensible, the “heterogeneity in data collection… limits the generalizability of findings.” This is a chief weakness and undercuts the assertions that are made throughout the rest of the chapter. One way around this would be to discuss the articles in groups by author. However, even then, ll. 329-33 present a problem for making any claims in such a tiny study. The best that can be set is that the study, small as it is, suggests possible…..

ll. 365-66: same issue here with assertions based on extremely limited data

ll. 409-10: how is this clear?

ll. 424: where could this guidance come from?

ll. 432: Causation is not possible in these studies. PBL can enhance (as this piece points out on l. 455), or be an opportunity for learning.

ll. 437: Best to avoid the use of “trials,” usually limited to clinical studies.

l. 445: research gaps are not “filled up”

l. 447: no way to guarantee this.

Table 2: confusing; suggest adding columns or spaces to indicate that the same articles are categorized in multiple ways

**Do you want your identity to be public for this peer review?** For information about this choice, including consent withdrawal, please see our Privacy Policy

Reviewer #1: No

Reviewer #2: No

Reviewer #3: No

---

## [Author Response · Author response to Decision Letter 1]

20 Jun 2025

We appreciate the review comments, and we have duly addressed them now. We are also grateful to the editor for their professional guidance.

---

## [Decision Letter · Decision Letter 1]

22 Sep 2025

Dear Dr. Abugri,

Please explain in more depth how and why the initial 20,789 papers were excluded (line 245).

"The studies were also extremely variable in design, sample size, and assessment tools, precluding synthesis and resulting in an inability to conduct thematic or inferential statistical analysis" (lines 285-287). If there is no ability to synthesize or analyze, then what can we learn from this effort? I would suggest either expanding your study scope and subsequently your search criteria to include more papers, or find a way to analyze the papers you did include qualitatively or thematically. For ideas, you can view the following, though there are other sources you can use:

Braun, V., & Clarke, V. (2006). Using thematic analysis in psychology. Qualitative research in psychology, 3(2), 77-101.

Conducting such an analysis would require you to add to your introduction and return to those sources in the findings, making your paper richer and more insightful for readers.

We look forward to receiving your revised manuscript.

Kind regards,

Rea Lavi

Academic Editor

PLOS ONE

Journal Requirements:

Additional Editor Comments:

Reviewer #4:

Reviewer #5:

Reviewers' comments:

Reviewer's Responses to Questions

**Comments to the Author**

Reviewer #4: (No Response)

Reviewer #5: (No Response)

2. Is the manuscript technically sound, and do the data support the conclusions?

Reviewer #4: Partly

Reviewer #5: Partly

3. Has the statistical analysis been performed appropriately and rigorously?

Reviewer #4: N/A

Reviewer #5: Yes

4. Have the authors made all data underlying the findings in their manuscript fully available?

Reviewer #4: Yes

Reviewer #5: Yes

5. Is the manuscript presented in an intelligible fashion and written in standard English?

Reviewer #4: Yes

Reviewer #5: Yes

Reviewer #4: I acknowledge that this is an ambitious study mapping available evidence on PBL effects. However, I believe the manuscript is incomplete for the following reasons:

1) As the authors state in their limitations, "The overall number of studies that were included was small (n = 8), and only one of these focused specifically on first-year medical students, which lowers the generalizability and degree of analysis." Additionally, while "20,789 studies were excluded during this rigorous process," the specific reasons for exclusion at each stage remain unclear, making it difficult to assess the comprehensiveness of the review.

2) The authors acknowledge that "Most of the studies employed self-reported data that may yield subjective bias and undermine the validity of the findings" and used "cross-sectional designs, which made it harder to prove causative relationship or long-term effects." Furthermore, "The heterogeneity in data collection methods complicates direct comparisons across studies and limits the generalizability of findings." These methodological variations, particularly in measuring critical thinking skills, make it challenging to synthesize findings meaningfully.

3) While the authors state this is "a scoping review [that] maps available evidence," the abstract and conclusion make claims about PBL "may support the development of critical thinking and communication skills" and suggest "effectiveness," which goes beyond the stated objective of identifying "key themes and gaps in the literature."

4) Although "data description was summarized descriptively, using mainly thematic trends, and contextual comparison between resource-rich and poor settings," there is limited in-depth thematic analysis. The influence of "cultural backgrounds" and "diverse educational backgrounds" mentioned by the authors is not systematically examined. Consequently, while the authors aim to "optimize PBL's implementation," it remains unclear how medical educators can practically apply these findings in their specific contexts.

Reviewer #5: The manuscript explores the effects of problem-based learning (PBL) on undergraduate medical students' critical thinking and communication skills development using a scoping review framework proposed by Arksey and O'Malley. The set of skills selected is relevant to the medical education field.

Comments:

1. The title and abstract reflect the title of the study. Suggest to revise the keyword first year medical student, as it is also linked to medical education another one of your keyword.

2. The introduction provides a context for the study, however, it is long-winded. There seems to be a disconnect between the history of PBL and twenty-first-century education. It is suggested to bridge the gap by emphasizing the core problem faced by the medical students in general, leading to the first year undergraduate students. Is there a gap between the PBL offered in resource-rich and resource-poor areas, is this connected to the availability of technology?

3. The definition for critical thinking needs and communications skills need to be clarified as the text indicated other the expansion of variables such as critical thinking is represented by (analytical reasoning and problem-solving) and communication skills is represented by (teamwork and interpersonal communication)

4. The methodology was satisfactorily explained. The PRISMA flow diagram was included.

5. Table 1: Data extraction text is unclear.

6. For the results section, there happens to be a conflict on generic skills (Problem-solving and collaborative skills, when it was mentioned as generic skills are (Problem-solving and analytical reasoning) in the inclusion criteria.

7. The discussion was rather limited to outdated citations; to enhance the write-up by using current literature.

8. Conclusion sums up the study’s scope well.

**Do you want your identity to be public for this peer review?** For information about this choice, including consent withdrawal, please see our Privacy Policy

Reviewer #4: No

Reviewer #5: No

---

## [Author Response · Author response to Decision Letter 2]

13 Oct 2025

Below, we provide a point-by-point response to each of the comments raised. All changes in the manuscript have been highlighted using the "Track Changes" feature for your convenience.

Response to Editor's Comments

Comment 1: "Please explain in more depth how and why the initial 20,789 papers were excluded"

Response: Thank you for this important request for methodological transparency. We have now added a detailed paragraph in the Results section that explicitly outlines the reasons for exclusion at both the title/abstract screening stage and the full-text review stage. We acknowledge that this value 20,789 was mistakenly recorded instead of 212 and we have now corrected it. This provides a clear, step-by-step account of the screening process, including the primary reasons for exclusion (e.g., wrong population, intervention, outcome, publication type). This addition can be found on page [10-11], and lines [211-232].

Comment 2: "If there is no ability to synthesize or analyze, then what can we learn from this effort? I would suggest... find a way to analyze the papers you did include qualitatively or thematically... Braun, V., & Clarke, V. (2006)."

Response

We appreciate this valuable suggestion. Our original wording may have unintentionally implied that we did not conduct any synthesis. We have now explicitly described the descriptive thematic synthesis of the included studies, drawing on the framework proposed by Braun and Clarke (2006) and Nowell et al. (2017). This is detailed in the Methods section under subheading Synthesis of results on page (10) and lines (200-207) with a new section titled thematic synthesis of result added. To clarify, while heterogeneity in study design, sample size, and assessment tools precluded quantitative or inferential statistical synthesis, this is consistent with the aims of a scoping review (Arksey & O’Malley, 2005; Tricco et al., 2018). We have revised the manuscript to reflect this.

Response to Reviewer #4's Comments

We thank Reviewer #4 for their thoughtful engagement with our manuscript.

Comment 1: "The specific reasons for exclusion at each stage remain unclear, making it difficult to assess the comprehensiveness of the review."

Response: We agree that this was a critical point to clarify. We have now added a detailed description in the results section that explicitly outlines the reasons for exclusion at both the title/abstract screening stage and the full-text review stage. Also, we have revised the PRISMA diagram appropriately to reflect this new addition. This provides a clear, step-by-step account of the screening process, including the primary reasons for exclusion. This addition can be found on pages [11-12], lines [211-232].

Comment 2: "These methodological variations... make it challenging to synthesize findings meaningfully."

Response: We acknowledge the inherent methodological limitations of the primary studies. Our adoption of a thematic analysis was specifically chosen to address this challenge. By synthesizing the data thematically rather than statistically, we were able to identify meaningful patterns and insights across heterogeneous studies. We have also been more explicit in acknowledging these limitations in the Discussion section, framing our findings as tentative but valuable pointers for future research.

Comment 3: "The abstract and conclusion make claims about PBL 'may support the development...' and suggest 'effectiveness,' which goes beyond the stated objective..."

Response: Thank you for this important correction. We have carefully revised the language throughout the manuscript, particularly in the Abstract and Conclusion, to ensure our claims are appropriately cautious and consistent with the exploratory nature of a scoping review. We now use phrases such as "may be associated with perceived improvements" and "findings remain tentative," which more accurately reflect our findings.

Comment 4: "...limited in-depth thematic analysis... influence of 'cultural backgrounds'... is not systematically examined... unclear how medical educators can practically apply these findings."

Response: This was an excellent point, and we have made significant additions to address it. First, we have added a new subsection in the Discussion titled "Influence of cultural and diverse educational backgrounds” to systematically examine this crucial factor see pages (23-24) and lines (455-475). Second, and most importantly, we have added a dedicated major section titled "Practical implications for educators implementing PBL in different resource settings see pages (26-27) and lines (517-540)." This new section translates our findings into actionable advice for educators, providing specific guidance for resource-rich, resource-limited, and culturally diverse contexts.

Response to Reviewer #5's Comments

We thank Reviewer #5 for their valuable suggestions, which have helped us refine the manuscript.

Comment 1: "Suggest to revise the keyword first year medical student..."

Response: We have revised the keyword list as suggested to include "first year medical student" for better indexing.

Comment 2: "The introduction... is long-winded... a disconnect between the history of PBL and twenty-first-century education."

Response: We have streamlined the Introduction to create a more direct narrative link between the historical context of medical education and the contemporary challenges faced by first-year students in different resource settings, thereby better framing the rationale for our study.

Comment 3: "The definition for critical thinking needs and communications skills need to be clarified..."

Response: We have clarified the operational definitions of both critical thinking and communication skills in the Introduction, explicitly linking them to their measurable indicators (e.g., analytical reasoning, problem-solving and collaborative skills) to ensure conceptual precision at Pages (4-5) and lines (87-95).

Comment 5: "Table 1: Data extraction text is unclear."

Response: We have reformatted Table 1 to improve its clarity and readability, ensuring the data is presented in a professional and unambiguous manner.

Comment 6: "For the results section, there happens to be a conflict on generic skills..."

Response: We have standardized the terminology for "generic skills" throughout the manuscript (inclusion criteria, Table 1, and main text) to resolve the noted inconsistency. The definition is now consistently presented as “(analytical reasoning, problem-solving and collaborative skills)."

Comment 7: "The discussion was rather limited to outdated citations..."

Response: We have updated the Discussion, particularly in the newly added sections, with more current literature (including citations from 2022-2024) to ensure our arguments are situated in the most recent scholarly context. Also, the scoping review was conducted in 2024, and inclusion criteria included studies from 2015 to 2024, hence the discussion featured studies from 2015 upwards.

---

## [Decision Letter · Decision Letter 2]

8 Jan 2026

Dear Dr. Abugri,

Thank you for submitting your manuscript to PLOS ONE. After careful consideration, we feel that it has merit but does not fully meet PLOS ONE’s publication criteria as it currently stands. Therefore, we invite you to submit a revised version of the manuscript that addresses the points raised during the review process.

We look forward to receiving your revised manuscript.

Kind regards,

Rea Lavi

Academic Editor

PLOS One

Journal Requirements:

Additional Editor Comments:

Dear Authors,

**Comments to the Author**

Reviewer #5: All comments have been addressed

Reviewer #6: (No Response)

Reviewer #7: All comments have been addressed

2. Is the manuscript technically sound, and do the data support the conclusions?

Reviewer #5: Yes

Reviewer #6: No

Reviewer #7: Yes

3. Has the statistical analysis been performed appropriately and rigorously?

Reviewer #5: Yes

Reviewer #6: N/A

Reviewer #7: N/A

4. Have the authors made all data underlying the findings in their manuscript fully available?

Reviewer #5: Yes

Reviewer #6: Yes

Reviewer #7: Yes

5. Is the manuscript presented in an intelligible fashion and written in standard English?

Reviewer #5: Yes

Reviewer #6: Yes

Reviewer #7: Yes

Reviewer #5: (No Response)

Reviewer #6: thankyou for undertaking this review. Unfortunately I have two specific issues in relation to the methods which make the manuscript unsuitable for publication at this time. - specifically a comprehensive search strategy was not used therefore limiting studies identified

Firstly the authors have only used 'problem-based learning' in the search - this excludes 2 important synonyms : case-based learning and inquiry based learning

Secondly - ERIC was not included. While I would agree studies are likely to be in PubMed or Proquest, ERIC is the comprehensive education data base.

Reviewer #7: The authors have responded to reviewers' comments and have updated the manuscript. A few additional suggestions are below:

• Fix the in-text citation style. Remove the comma before numbering at the end of each sentence.

• Several citations are too old (For example, 1986). Please provide updated citations.

• Elaborate on resource-rich and resource-limited settings. What resources are needed for PBL? Give examples of PBL conducted in resource-rich and resource-limited settings.

**Do you want your identity to be public for this peer review?** For information about this choice, including consent withdrawal, please see our Privacy Policy

Reviewer #5: No

Reviewer #6: No

Reviewer #7: No

---

## [Editor Report · Decision Letter 3]

26 Jan 2026

The effects of problem-based learning (PBL) on undergraduate medical students' critical thinking and communication skills development. A scoping review across resource-rich and resource-limited settings (2015-2024).

PONE-D-25-14527R3

Dear Dr. Abugri,

We’re pleased to inform you that your manuscript has been judged scientifically suitable for publication and will be formally accepted for publication once it meets all outstanding technical requirements.

Kind regards,

Rea Lavi

Academic Editor

PLOS One

Additional Editor Comments (optional):

Please shorten the title of the manuscript so that it does not take up more than two lines.

Please remove ".0" from section titles. Section titles should be, for example, "1. Introduction".

---

## [Editor Report · Acceptance letter]

PONE-D-25-14527R3

PLOS One

Dear Dr. Abugri,

I'm pleased to inform you that your manuscript has been deemed suitable for publication in PLOS One. Congratulations! Your manuscript is now being handed over to our production team.

Kind regards,

on behalf of

Dr. Rea Lavi

Academic Editor

PLOS One